# Colloidal Processing of Complex-Shaped ZrB_2_-Based Ultra-High-Temperature Ceramics: Progress and Prospects

**DOI:** 10.3390/ma15082886

**Published:** 2022-04-14

**Authors:** Guoqian Liu, Changhai Yan, Hua Jin

**Affiliations:** 1Science and Technology on Space Physics Laboratory, China Academy of Launch Vehicle Technology, Beijing 100076, China; 15201631565@163.com (G.L.); yanchanghai@163.com (C.Y.); 2School of Aerospace Engineering, Xiamen University, Xiamen 361005, China

**Keywords:** ultra-high-temperature ceramics, complex shapes, colloidal processing, additive manufacturing

## Abstract

Ultra-high-temperature ceramics (UHTCs), such as ZrB_2_-based ceramics, are the most promising candidates for ultra-high-temperature applications. Due to their strong covalent bonding and low self-diffusion, ZrB_2_-based UHTCs are always hot-pressed at temperatures above 1800 °C. However, the hot-pressing technique typically produces disks or cylindrical objects limiting to relatively simple geometrical and moderate sizes. Fabrication of complex-shaped ZrB_2_-based UHTC components requires colloidal techniques. This study reviews the suspension dispersion and colloidal processing of ZrB_2_-based UHTCs. The most important issues during the colloidal processing of ZrB_2_-based UHTCs are summarized, and an evaluation of colloidal processing methods of the ZrB_2_-based UHTCs is provided. Gel-casting, a net or near-net colloidal processing technique, is believed to exhibit a great potential for the large-scale industrialization of ZrB_2_-based UHTCs. In addition, additive manufacturing, also known as 3D printing, which has been drawing great attention recently, has a great potential in the manufacturing of ZrB_2_-based UHTC components in the future.

## 1. Introduction

Ultra-high-temperature ceramics (UHTCs) are a family of materials that include a number of borides, carbides, and nitrides of the group IVB and VB transition metals, with a significant criterion that their melting temperatures are in excess of 3000 °C [1,2,3,4,5]. In recent years, zirconium diboride- (ZrB_2_) [6,7,8,9,10] and hafnium diboride (HfB_2_)-based [11,12,13,14,15] UHTCs have attracted increasing attention from material scientists and engineers due to their superb combination properties of high melting point, high strength and hardness, high electrical and thermal conductivities, good chemical inertness, excellent corrosion resistance, and good oxidation resistance. Usually, ZrB_2_-based UHTCs are designed to be potential materials for use in extreme environments, such as scramjet engine components, sharp leading edges, nosecones and thermal protection systems for reusable re-entry vehicles, hypersonic flights, and rocket propulsion systems [16,17,18,19,20].

However, there is an inherent drawback in the fabrication of ZrB_2_-based UHTC components. Due to their strong covalent bonding and low self-diffusion, in most conditions, UHTCs need to be hot-pressed at temperatures above 1800 °C, depending on the sintering parameters (pressure, soaking time, heating rate) and/or sintering additives [21,22,23,24,25,26,27,28,29,30]. However, the hot-pressing technique typically produces disks or cylindrical objects, limiting to relatively simple geometrical and moderate sizes. Fabrication of complex-shaped components, for example, the sharp leading edges in hypersonic vehicles, usually requires post-machining treatments. However, these conventional diamond machining or electrical discharge machining treatments are usually costly. Besides, these post-machining treatments also increase the failure risks of cracks and defects in the final component [31,32,33].

Fortunately, colloidal processing is an alternative route for the processing of complex-shaped ZrB_2_-based UHTCs, which has gained great attention in the past decades. As is known, the colloidal processing technologies for ceramics are usually based on the control of forces between particles in suspensions. By controlling the dispersion and stabilization of ceramic particles among a specific solvent or suspension, the agglomerates between ceramic particles can be broken down, and an intimate mixing degree between different compounds can be obtained, which can minimize the possibility of defects in the final material [34,35,36,37]. The colloidal processing of ceramics is mainly composed of two significant parts: dispersion of ceramic particles and the colloidal processing procedure. Consequently, a number of reports have therefore recently been focused on the colloidal processing of ZrB_2_-based UHTCs.

Herein, we reviewed the suspension dispersion and colloidal processing of ZrB_2_-based UHTCs. The purpose of this review is to summarize the most important issues during the colloidal processing of ZrB_2_-based UHTCs and to provide an evaluation of colloidal processing studies of the ZrB_2_-based UHTCs. Besides, the prospects for the fabrication of complex-shaped ZrB_2_-based UHTC components, such as additive manufacturing (3D printing), were also forecasted.

## 2. Dispersion of ZrB_2_ Powder

Usually, for the colloidal processing of ZrB_2_-based ultra-high-temperature ceramics (UHTCs), there are two kinds of colloidal processing routes, including aqueous colloidal processing and non-colloidal processing. Normally, non-colloidal processing of ceramics requires the use of polymer or chemical mediums, which usually have some toxicity and pollution problems for the environment. Therefore, aqueous colloidal processing, which has advantages such as being non-toxic, non-polluting, and low cost, has drawn great attention for the fabrication of complex-shaped ceramics. Hence, here, we mainly focus on the issues during the aqueous colloidal processing of ZrB_2_-based UHTCs.

### 2.1. Corrosion and Hydrolysis of ZrB_2_ Powders

During the colloidal fabrication of ZrB_2_-based UHTC products, wet processes, especially aqueous colloidal processing, such as ball-milling, mixing, and suspension preparation, are usually required. Corrosion and hydrolysis could happen on the surface of ZrB_2_ particles during this colloidal processing. The corrosion behaviors of ceramic powders such as B, B_4_C, and BN in water have been widely reported, and these studies have shown that the particles are quickly oxidized after manufacture and aqueous colloidal processing, and become coated with a thin layer of oxygen impurities containing boric acid (H_3_BO_3_) and boron oxide (B_2_O_3_). ZrB_2_ particles are usually oxidized in air after manufacture and the particle surface is oxidized to ZrO_2_ and B_2_O_3_, as indicated below:(1)$$ZrB_{2}+\frac{5}{2}O_{2}=ZrO_{2}+B_{2}O_{3}$$

Besides, during the wet processes, chemical bonding on the surface of ZrB_2_ particles usually changes and oxygen impurities are always introduced, as shown in Reactions (2)–(4). It is widely accepted that the surface properties are very crucial for the following colloidal processing, and the oxygen content also plays a significant negative role in the sintering and densification process of ZrB_2_-based UHTCs. As a consequence, controlling the corrosion and hydrolysis behavior of ZrB_2_ powders in aqueous medium is of paramount importance during the colloidal processing.
(2)$$ZrO_{2}+2H_{2}O=Zr{\left(OH\right)}_{4}$$
(3)$$B_{2}O_{3}+3H_{2}O=2H_{3}BO_{3}$$
(4)$$ZrB_{2}+10H_{2}O=Zr{\left(OH\right)}_{4}+2H_{3}BO_{3}+5H_{2}↑$$

Lee et al. [38] reported the corrosion behavior of ZrB_2_ powders in water during static aging and dynamic stirring. It was found that the corrosion of ZrB_2_ powders did not intensively occur at pH values of 3–11 in static water. However, the corrosion was significantly enhanced when stirring the ZrB_2_ suspensions. They found that the surface of the ZrB_2_ particles was mainly covered with Zr-OH and Zr-B bonding. Yin et al. [39] also investigated the hydrolysis behavior of ZrB_2_ powders. It was found that the hydrolysis level of ZrB_2_ powders increased as the ball-milling speed or time increased, and the surface of the particle was mainly covered with an oxide layer composed of Zr-OH and B-OH, with a thickness of about 5 nm, which was believed to act as a hydrophilic coating and help the dispersion of ZrB_2_ powders in dilute aqueous solution. The corrosion and hydrolysis mechanism is diagrammatically drawn in Figure 1, and the dimension does not represent the real particle size, nor do the numbers of Zr-OH and B-OH bonds indicate the real content ratio.

### 2.2. Dispersion Mechanism

During the colloidal fabrication of ZrB_2_-based UHTC products, wet processes, especially aqueous colloidal processing, such as ball-milling, mixing, and suspension preparation, dispersion is the most important issue for the following processing.

The most common feature for colloidal processing is small particles (micron-sized or nano-sized) with a large specific surface area (always above 10 m^2^/cm^3^). During the suspension fabrication, the long-range Van der Waals force is ubiquitous and significant, and all ceramic particles tend to attract each other and form an aggregate. Therefore, one or more kinds of repulsive forces should be introduced between ceramic particles in order to tailor the suspension stability through electrostatic interaction, steric, or other repulsive interactions. Typically, a polymer dispersant is usually added to the colloidal system to enhance the electrostatic and steric repulsive force in order to prevent aggregation. Commonly, electrostatic and steric stabilization are used to stabilize ceramic particles in an aqueous system.

The interparticle interactions between ceramic particles can be calculated semi-quantitatively. The interparticle interactions are evaluated by using the Derjaguin–Landau–Verwey–Overbeck (DLVO) model. The total interparticle interaction (*V*) can be described as follows [40,41]:(5)$$V=V_{\mathrm{VdW}}+V_{\mathrm{electrosta}t}+V_{\mathrm{steric}}$$
where *V*_VdW_, *V*_electrostat_*,* and *V*_steric_ are the attractive Van der Waals, the electrostatic, and the steric interactions, respectively. These interactions will be briefly introduced in the following sections.

#### 2.2.1. Van der Waals Force

Among a ceramic suspension system, *V*_VdW_ can be calculated by the Hamaker model as [40,41]:(6)$$V_{\mathrm{VdW}}=\frac{A}{6}\left[\frac{2a_{1}a_{2}}{d^{2}+2a_{1}d+2a_{2}d}+\frac{2a_{1}a_{2}}{d^{2}+2a_{1}d+2a_{2}d+4a_{1}a_{2}}+\ln\left(\frac{d^{2}+2a_{1}d+2a_{2}d}{d^{2}+2a_{1}d+2a_{2}d+4a_{1}a_{2}}\right)\right]$$
where *A* is the effective Hamaker constant for the system, *d* is the particle surface–surface separation, and *a*_1_ and *a*_2_ are the ceramic particle radii, respectively. If the suspension system is a mono-particle system, which means there is only one kind ceramic particle, *a*_1_
*= a*_2_ in this condition.

#### 2.2.2. Electrostatic Interaction

By generating like-charges of sufficient magnitude on the surfaces of suspended ceramic particles, the stability of the aqueous suspension can be controlled. The as-resulted repulsive potential energy, *V*_electrostat_, usually exhibits exponential distance dependence. The strength of *V*_electrostat_ strongly depends on the surface potential induced on the interacting colloidal particles and the dielectric properties of the intervening medium. As is known, *V*_electrostat_ is usually considered based on the Hogg–Healy–Füstenau (HHF) model. In this model, the dispersion medium can be characterized by the ionic strength, *I*_c_, as provided by Equation (7) [40,41]:(7)$$I_{c}=\frac{1}{2}\sum c_{i}z_{i}{}^{2}$$
where *c_i_* is the molar concentration of the ionic species *i*, having a valence *z_i_*. Taking the ZrB_2_ particle as an example, the electrostatic potential for the ZrB_2_ particle obeys the interacting double-layer model. The electrical charges of the ZrB_2_ particle can be achieved by the preferential surface hydrolysis and dissolution of the ceramic particle. Figure 2 shows the double-layer structure of a positive surface-charged ZrB_2_ particle. In this double-charge layer structure of the ZrB_2_ particle, as shown in Figure 2, stern potential reflects the particle charge, but it is nearly impossible to be measured or calculated from the experiment, while Zeta potential is a shear plan potential, and is always used as a criterion for surface charge because it is close to the stern potential and it can be measured [40,41].

The double-charge layer formed on the surface of the ZrB_2_ particle can be calculated by using the Debye length (double-layer thickness), *κ*^−1^, as provided by Equation (8) [40,41]:(8)$$\kappa^{-1}=\sqrt{\frac{\varepsilon_{m}\varepsilon_{e}kT}{2e^{2}I_{c}1000N_{A}}}$$
where *ε_m_* and *ε_e_* are the dielectric constant of the medium and the electric constant, respectively, *k* is Boltzmann’s constant, *T* is the absolute temperature, *e* is the elementary charge, and *N_A_* is Avogadro’s constant.

The surface potential, *Ψ**,* can be calculated from the as-measured Zeta potential (*ζ*) by using [40,41]:(9)$$\psi =\xi \exp(\kappa d_{s})$$

In this equation, *d_s_* is the distance from the surface where the Zeta potential can be measured.

Therefore, according to the HHF model, the electrostatic interaction, *V*_electrostat_, can be calculated as follows [40,41]:(10)$$V_{\mathrm{electrostat}}=\frac{\pi \varepsilon_{m}\varepsilon_{e}a_{1}a_{2}}{2a}\left[{\left(\psi_{1}+\psi_{2}\right)}^{2}\ln\left(1+\exp\left(-\kappa h\right)\right)+{\left(\psi_{1}-\psi_{2}\right)}^{2}\ln\left(1-\exp\left(-\kappa h\right)\right)\right]$$

Normally, Zeta potential means the electrostatic dispersion effect. The higher the relative Zeta potential, the better the dispersion behavior of an aqueous ceramic suspension. Zeta potential is usually affected by many factors, such as surface chemistry, surface charge, pH value, ionic concentration, and dispersant absorption. Ionic dispersants are usually added to enhance the dispersion and stability of the aqueous ceramic suspension. Firstly, the ionic dispersant acts as an electrolyte by increasing the ionic concentration in the medium. This affects the surface charge and Zeta potential of the dispersed particles. Zeta potential is related not only to the surface charge of the ceramic particles, but also to the ionic concentration and the dispersant content. In most conditions, the ionic concentration can also be controlled by the dispersant content. Therefore, the dispersant content becomes one of the most important factors that affect the Zeta potential. Secondly, the dispersant usually adsorbs onto the surface of a ceramic particle among an aqueous ceramic suspension. The adsorbed dispersant acts in various ways based on the amount added and the chain characteristics of the long chain of the polymer dispersant. The desired effect of adsorption is to produce a steric repulsive force and stabilize the ceramic suspension. Besides, in some conditions, the polymer dispersant can bridge between particles and form a loss-flocculated structure.

#### 2.2.3. Steric Interaction

Steric interaction also provides an alternative route to control the colloidal stability of an aqueous ceramic suspension that can be used in polar and non-polar medium. The mechanism for steric stabilization involves the presence of a lyophilic colloid, which adsorbs onto the surface of a ceramic particle and provides a steric force. Many theories for steric interaction mechanisms have been proposed and explained. The most generally accepted mechanism is the hard-wall model, which was proposed by Bergstrom. The interaction potential for this model can be divided into three domains, the first being a fully interpenetrated domain at separations closer than the adsorbed layer thickness, *d_a_*, where the interaction potential is infinitely repulsive. The interpenetrated domain up to a separation of twice the adsorbed layer thickness is characterized by the molecule–solvent interaction, *χ*, the molecular volume of the solvent, *V*_s_, as well as the volume fraction of molecules in the adsorbed layer, *φ*. At large separations, the adsorbed molecules do not interact, and the potential is consequently zero [40,41].
(11)$$V_{\mathrm{steric}}=\left\{\begin{array}{l} h<d_{a}:\infty \approx 10^{6}\\ d_{a}\leq h\leq 2d_{a}:\\ h>d_{a}:0 \end{array}\right.\frac{2a_{2}}{a_{1}+a_{2}}\left[\frac{\pi a_{1}\kappa T}{V\varphi^{2}}\left(\frac{1}{2}-\chi \right){\left(2d_{a}-h\right)}^{2}\right]$$

When two ceramic particles with adsorbed polymer layers approach each other at a distance of less than twice the thickness of the adsorbed layer, steric interaction occurs between the two layers. If the particles get close to each other, the particles will lose the degree of freedom and the entropy will decrease, and the system increases.

The steric stabilization is usually used for non-polarized ceramic dispersal. It can also combine with electrostatic stabilization to increase the stability of the colloidal system. When two ceramic particles are attracted, the electrostatic interaction takes place first because it has a longer effective distance than the steric interaction. When the particles get closer, the electrostatic and steric interactions of the polyelectrolyte work together. This kind of stabilization is achieved in suspensions by using polyelectrolytes, which contain at least one type of ionizable group with an additional molecular structure that ranges from homopolymers such as polyacrylic acid (PAA), polyethylene imine (PEI), etc.

He et al. [41] calculated the interparticle interactions between ZrB_2_ micron-sized particles during dispersion in aqueous medium using a special polyelectrolyte, branched polyethylenimine (PEI), as the dispersant (Figure 3). When adding polyelectrolytes, the attractive Van de Waals forces, electrostatic interaction, and steric interaction were coupled. It was found that the interparticle interactions between the ZrB_2_ particles exhibited a significant repulsive force when using a certain amount of dispersant.

Since the dispersion mechanisms of ZrB_2_ particles among suspension have been clearly investigated and figured out, the dispersion of ZrB_2_ ceramic slurries becomes possible.

### 2.3. Dispersion of ZrB_2_ Powders

Ionic and/or polyelectrolyte dispersants were used for the dispersion of ceramic particles, such as ZrB_2_ particles, in medium. Table 1 lists the achievements of the dispersion of micron-sized and nano-sized ZrB_2_ particles in water or organic solvent. In Table 1, the dispersant used, the optimum dispersant concentration, and the optimum pH value are summarized. Besides, the co-dispersion of ZrB_2_ particles with other additives, such as SiC, B_4_C, and C, was also investigated in the last decades. Based on these research achievements, well-dispersed ZrB_2_ suspensions were obtained, which established the prerequisite of the colloidal processing of ZrB_2_-based ceramics.

## 3. Colloidal Processing

### 3.1. Slip-Casting

During slip-casting, ceramic particles were consolidated into sophisticated shapes with high green density. Currently, it is used mainly for the production of large components or for those having these walls or a shell of complicated contours. Medri et al. [45,46] set up slip-casting of concentrated aqueous ZrB_2_-SiC composite suspensions as a forming technique for the production of UHTC crucibles. In this study, the effects of two commercial ammonium polyacrylates dispersants (Duramax D3005 and Dolapix PC33) on the dispersion and the final microstructures were investigated. ZrB_2_-SiC crucibles with an almost homogeneous microstructure and lower final porosity were finally prepared through slip-casting, as shown in Figure 4.

However, on one hand, the main disadvantage of this process is that it is time-intensive, with the speed of casting ranging from hours to days for large objects. Besides, there is little difference in microstructure near and far away from the plaster mold, owing to the difference of adsorption forces in different distances from the plaster mold. This difference finally results in density differences over the ceramic part. On the other hand, the slip-casting technique cannot produce net or near-net and complex-shaped materials. The dimensional accuracy of the slip-casted product is still relatively low.

### 3.2. Tape-Casting

Tape-casting is a widespread and cost-effective colloidal processing method that is mainly used to make thin and flat ceramic sheets. In the last two decades, the tape-casting technique has been widely used for the fabrication of oxide and non-oxide ceramics, such as Al_2_O_3_ [55,56], ZrO_2_ [57,58], SiC [59,60], Si_3_N_4_ [61], etc. Natividad et al. [50] fabricated flexible and homogeneous ZrB_2_ tapes with a thickness of about 280 μm from an organic solvent-based slurry using tape-casting, as shown in Figure 5. Lü [42,62] and Medri [63] et al. also successfully prepared ZrB_2_ and ZrB_2_-SiC tapes from suspensions using the tape-casting technique.

Moreover, tape-casting is also reported to build up multi-layered structures with complex geometries and improved properties, such as higher fracture toughness and better thermal shock behavior, which are extremely important for the engineering applications of UHTCs in aerospace. Laminated ZrB_2_ [64] and ZrB_2_-SiC [65] tapes were also obtained by the aqueous tape-casting process and subsequent hot-pressing. Wei et al. also fabricated laminated ZrB_2_-SiC/graphite [66] and ZrB_2_-SiC/BN [67] ceramics with simultaneously improved flexural strength, fracture toughness, and thermal shock resistance, as shown in Figure 6.

However, the main disadvantage of the tape-casting process is that it can only prepare thin-layer or laminated ceramics, and complex-shaped ceramic components cannot be obtained.

### 3.3. Gel-Casting

Gel-casting, which is a newly developed ceramic-forming technique, has gained great attention in the past decades. Janney and Omatete [68,69,70] first developed the gel-casting technique for complex-shaped ceramics. The gel-casting process is based on the casting of ceramic slurry, which contains ceramic powder’s water and water-soluble organic monomers. After casting the monomer, the mixture can be polymerized to form a gelled part. Drying, polymer burn-out, and sintering will be subsequently conducted, and the manufacturing route is then completed. The gelled ceramic body is removed from the mold while still wet, and then dried and fired. During the preparation of ceramic suspension, it is advantageous to utilize colloidal processing, which is of exceptional importance for reliable and defect-free processing of sub-micrometer- and nanometer-sized powders with their strong tendency to agglomeration. Moreover, gel-casting separates the shaping (i.e., the casting stage) of the suspension from the consolidation stage. During the consolidation stage, no movement of particles occurs, so the uniform particle dispersion of a stable suspension can be retained in the gelled ceramic body, which finally results in a highly regular particle packing in the green body. Therefore, the gel-casting process is generic and can be used for a wide range of ceramic powders, such as Al_2_O_3_ [68,69], ZrO_2_ [71,72], SiC [73,74], etc. In contrast with slip-casting, gelled ceramic parts are more homogeneous and have a much higher green strength, and gel-casting parts contain only a few percent of organic components, making binder removal much less critical compared to tape-casting and other colloidal processing methods [75].

Yin et al. [48] and He et al. [76,77,78,79] made a great deal of effort in the fabrication of ZrB_2_-based ceramics from aqueous suspensions using the aqueous gel-casting technique. Especially, He et al. [76,77] obtained complex-shaped ZrB_2_-SiC ultra-high-temperature ceramic components, as shown in Figure 7. It was found that the gel-casted ZrB_2_-SiC ceramic components had a homogenous microstructure and near-net shape in dimension. The traditional gel-casting procedure is usually based on the acrylamide [C_2_H_3_CONH_2_] (AM) and N,N0-methylenebisacrylamide [(C_2_H_3_CONH)_2_CH_2_] (MBAM) (AM-MBAM) system or the Na-alginate system, etc. In order to further improve the green density and green machinability of ZrB_2_-SiC ceramic components, He et al. [80] also developed a novel double-gel network-based gel-casting technique. In addition, ZrB_2_ and ZrB_2_-SiC ultra-high-temperature ceramic foams [81] and HfB_2_-SiC ultra-high-temperature ceramic [82] were also fabricated using the gel-casting method.

Moreover, other casting techniques, such as freeze-casting [83] and freeze-form extrusion fabrication [43], have also gained great attention recently. However, these casting techniques have not been widely used for the colloidal fabrication of UHTCs.

To sum up, gel-casting is a net shape-forming process similar to slip-casting and tape-casting, but with several advantages. Compared to slip-casting, gel-casting results in much more homogenous material with no density differences over the ceramic component. Compared to tape-casting, the gel-casting process is much more attractive. Gel-casting has the ability to produce not only thin-film but also complex-shaped ceramic components. Besides, the gel-casting usually uses only small quantities of organic binder, uses water as the suspension medium, does not include the critical binder removal step, and can also be used for prototypes and small series, such as for automated production. In conclusion, the gel-casting technique has the following advantages [75]:Capable of producing complex-shaped ceramic components.Low equipment and mold costs.Capable of mass production.High green density and green strength.Excellent green machinability.Very homogeneous microstructure and material properties.

Many enterprises have already been engaged in the production of ceramic parts in Al_2_O_3_ and ZrO_2_ on an industrial scale. The authors firmly believe that the gel-casting technique has a great potential for the large-scale industrialization of UHTCs in the near future.

## 4. Prospects

Additive manufacturing (AM), also called 3D printing, has been developing and has drawn great attention recently [84,85,86]. Compared with traditional manufacturing technologies (even traditional colloidal processing technologies), additive manufacturing cannot be restricted by mold-making or processing technology, solves the formation of complex-shaped structure products, and greatly reduces the processing procedures and shortens the processing cycle. Moreover, the more complex the product’s structure, the more significant the advantages of additive manufacturing. Great progress and breakthroughs have been made for the additive manufacturing of polymers [87,88,89,90] and metals [91,92,93,94,95] with complex geometries, and various ceramics with complex shapes have also been prepared by additive manufacturing.

Simply, there are many kinds of additive manufacturing technologies which have been developed recently for the fabrication of complex-shaped ceramics, including selective laser sintering (SLS), selective laser melting (SLM), binder jetting (BJ), fused deposition modeling (FDM), laminated object manufacturing (LOM), robocasting, extrusion-free forming (EFF), direct ink writing (DIW), stereolithography (SL), etc. [96].

Up to now, complex-shaped oxide ceramics (Al_2_O_3_ [97,98], ZrO_2_ [99,100], etc.), typical non-oxide ceramics (SiC [101,102,103], Si_3_N_4_ [104,105], etc.), and precursor-derived ceramics (SiOC [106,107], SiCN [108,109], etc.) have been reported to be successfully prepared by using different additive manufacturing technologies. Actually, each additive manufacturing technology has its own advantages and forming accuracy [96], as illustrated in Table 2. However, it was also found that UHTCs have rarely been prepared by additive manufacturing technologies. Fortunately, some attempts were carried out on the additive manufacturing of UHTCs, such as by Feilden et al. [110], who prepared HfB_2_-based UHTC by a so-called robocasting additive manufacturing technique, as shown in Figure 8a. And Kemp et al. [111], who also prepared ZrB_2_-SiC UHTC chopped fiber ceramic composites by a direct ink writing additive manufacturing technique, as shown in Figure 8b. To the best of our knowledge, the key issues involved in the additive manufacturing of UHTCs include the dispersion of a ceramic slurry, the processing of the green part, and the sintering, which are similar to traditional colloidal processing technologies. Therefore, it is believed that additive manufacturing techniques will be the most useful methods for fabricating complex-shaped UHTC components in the future.

## 5. Summary

ZrB_2_-based ultra-high-temperature ceramics (UHTCs) have drawn great attention for ultra-high-temperature applications. The colloidal processing technique is a promising route for net or near-net fabrication of complex-shaped ZrB_2_-based UHTC components. This paper reviewed the dispersion and colloidal processing of ZrB_2_-based UHTCs. The concluding remarks are listed as follows:(1)The corrosion and hydrolysis mechanisms of ZrB_2_ particles in aqueous medium have been summarized. The surface of the ZrB_2_ ceramic particle is mainly covered with an oxide layer composed of Zr-OH and B-OH, which should be controlled during the wet processes, especially aqueous colloidal processing, such as ball-milling, mixing, and suspension preparation.(2)The dispersion mechanism and dispersion method of ZrB_2_ ceramic particles (micron-sized and nano-sized) in aqueous medium have been summarized. From the researchers’ great efforts, well-dispersed ZrB_2_ ceramic suspensions can be obtained, which is in favor of the following colloidal processing.(3)Various colloidal processing methods, such as slip-casting, tape-casting, freeze-casting, and gel-casting, etc., have been introduced and evaluated. The advantages and disadvantages for each processing method have been summarized, and we provided an evaluation of colloidal processing studies of the ZrB_2_-based UHTC.

Gel-casting is the most promising net shape-forming process of ZrB_2_-based UHTC. We firmly believe that the gel-casting technique has a great potential for the large-scale industrialization of UHTCs. Besides, additive manufacturing, also known as 3D printing, which has been drawing great attention recently, is believed to have a great potential in the manufacturing of ZrB_2_-based UHTC components in the future.

## Figures and Tables

**Figure 1 materials-15-02886-f001:**
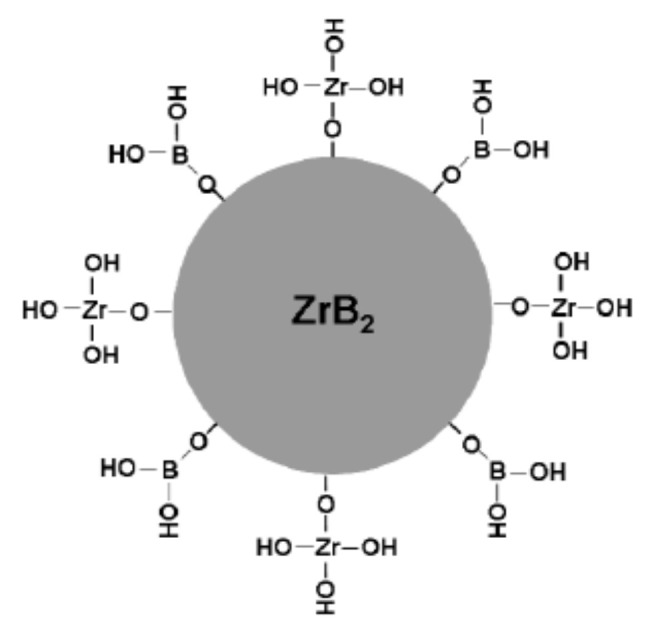
Corrosion and hydrolysis of the ZrB_2_ particle in water.

**Figure 2 materials-15-02886-f002:**
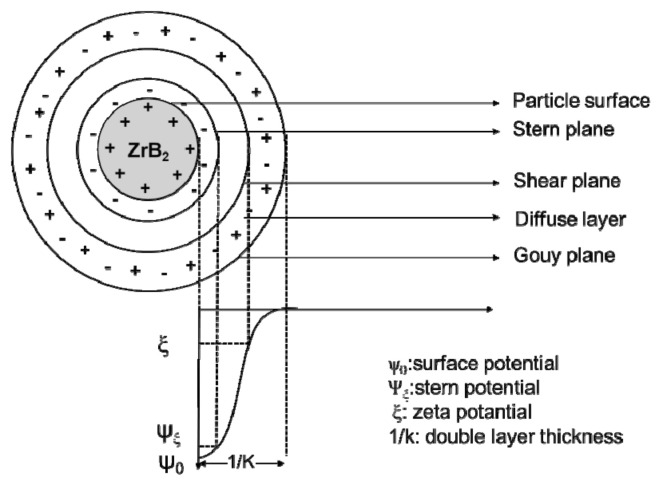
Schematic drawing of the double-charge layer structure of the ZrB_2_ particle.

**Figure 3 materials-15-02886-f003:**
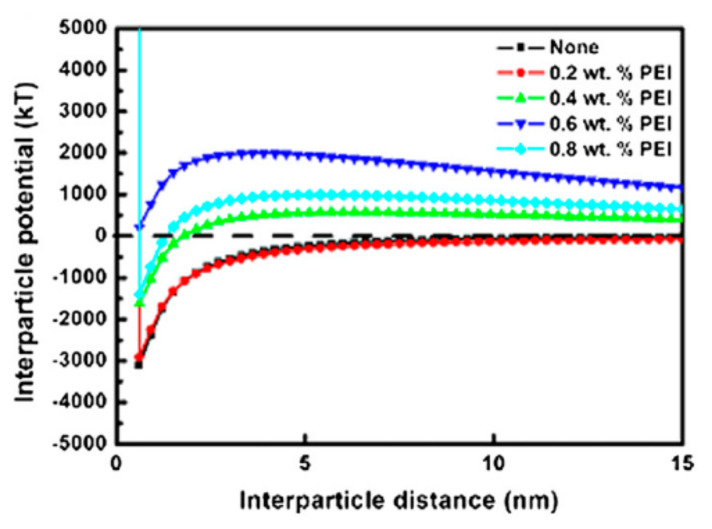
Interparticle interaction calculations for ZrB_2_ particles in aqueous medium. Reproduced with permission from [41].

**Figure 4 materials-15-02886-f004:**
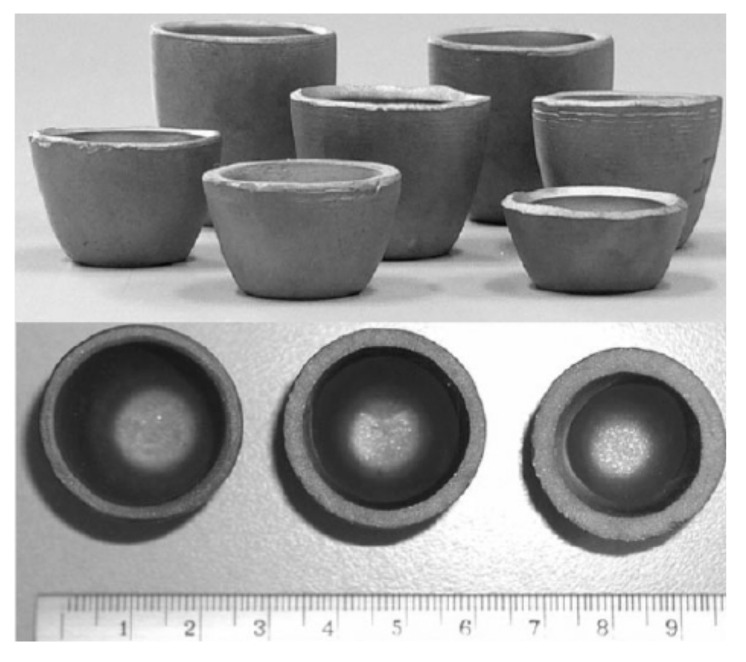
ZrB_2_-SiC ultra-high-temperature ceramic crucibles prepared by slip-casting. Reproduced with permission from [46].

**Figure 5 materials-15-02886-f005:**
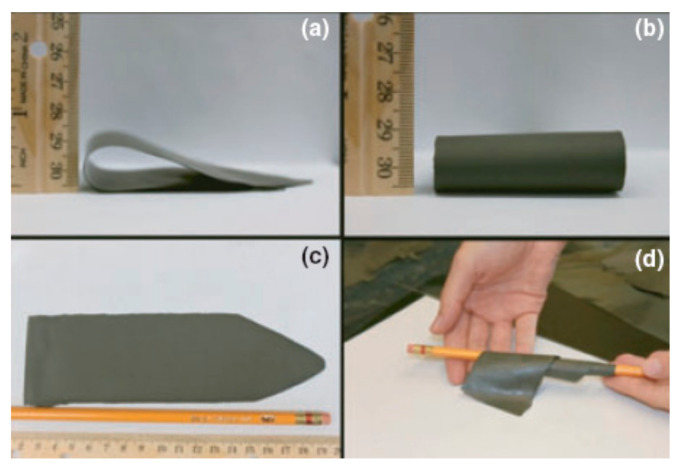
ZrB_2_ green tapes: (**a**) side view bend, (**b**) front view bend, (**c**) top view, and (**d**) wrapped around a pencil as a demonstration of the tape’s flexibility. Reproduced with permission from [50].

**Figure 6 materials-15-02886-f006:**
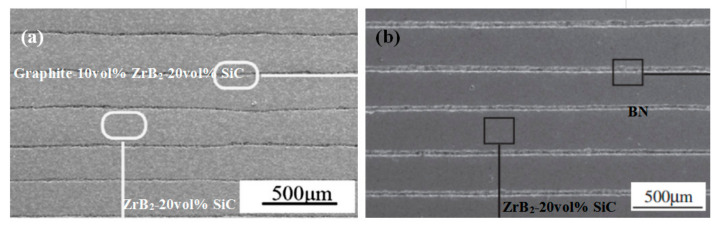
(**a**) Laminated ZrB_2_-SiC/graphite ceramic produced by tape-casting. Reproduced with permission from [66]. (**b**) ZrB_2_-SiC/BN ceramic produced by tape-casting. Reproduced with permission from [67].

**Figure 7 materials-15-02886-f007:**
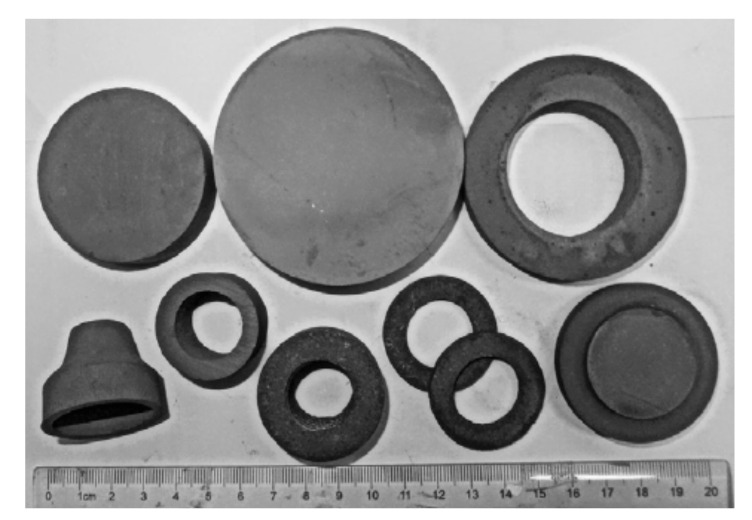
ZrB_2_-SiC ultra-high-temperature ceramic components. Reproduced with permission from [77].

**Figure 8 materials-15-02886-f008:**
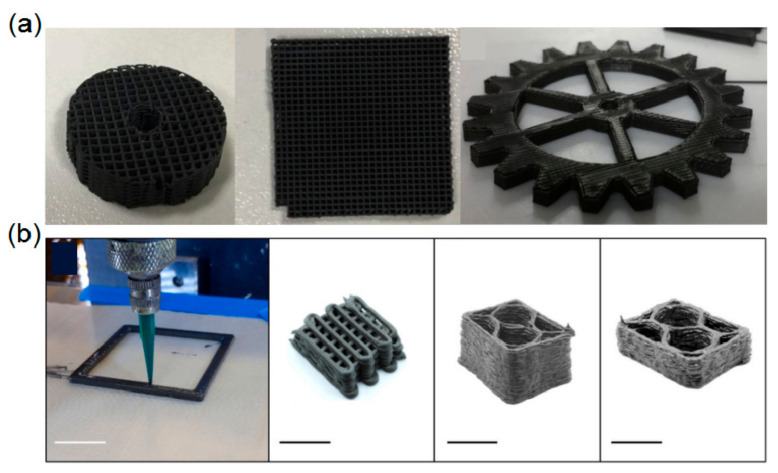
(**a**) Robocasting of HfB_2_-based UHTC. Reproduced with permission from [110]. (**b**) Direct ink writing of ZrB_2_-SiC UHTC chopped fiber ceramic composites, all scale bars were 1 cm. Reproduced with permission from [111].

**Table 1 materials-15-02886-t001:** Summary of the dispersion of the ZrB_2_ particle.

Powder	Dispersant	Optimum Dispersant Concentration	Optimum pH Value	Medium	Ref.
Micron-sized ZrB_2_	Lopon 885	0.4 wt.% ^a^	9	Water	[42]
	Zetasperse 1200	4.0 mg/m^2 b^	10	Water	[43]
	Dynol607	4.0 mg/m^2^	10	Water	[43]
	WA1	2.6 mg/m^2^	11	Water	[43]
	Darvan821A	1.25 mg/m^2^	9	Water	[43]
	PEI10000	1.5 wt.%	11	Water	[44]
	Duramax D3005	2.9 wt.%	6–7	Water	[45]
	Dolapix PC33	1.5–2.5 wt.%	9–10	Water	[46]
	Ammonium Citrate Tribasic	0.5 wt.%	9–10	Water	[47]
	SD-07	0.60 mg/m^2^	7–12	Water	[48]
	PEI	–	8	Water	[41,49]
	Blown Menhaden Fish Oil	1 wt.%	–	Organic solvent	[50]
Nano-sized ZrB_2_	PAA3000	1 wt.%	10	Water	[51]
	PEI10000	0.7 wt.%	10	Ethanol	[52]
	Gallic acid	6 wt.%	–	n-butanol	[53,54]

^a^ The weight ratio to the total powder weight. ^b^ The weight ratio to the specific surface area.

**Table 2 materials-15-02886-t002:** Various additive manufacturing technologies for complex-shaped ceramics [96].

Additive Manufacturing Technology	Forming Accuracy
Selective laser sintering (SLS)	μm–mm
Selective laser melting (SLM)	μm–mm
Binder jetting (BJ)	μm–mm
Fused deposition modeling (FDM)	mm
Laminated object manufacturing (LOM)	mm
Robocasting	μm–mm
Extrusion-free forming (EFF)	μm–mm
Direct ink writing (DIW)	μm–mm
Stereolithography (SL)	nm–μm

## Data Availability

No new data were created or analyzed in this study. Data sharing is not applicable to this article.

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
