# Peer review of "Colloidal Processing of Complex-Shaped ZrB2-Based Ultra-High-Temperature Ceramics: Progress and Prospects"

_materials, 2022, doi:10.3390/ma15082886_

Round 1

Reviewer 1 Report

The manuscript seems to be a useful review of a special process to produce ZrB2-based ultra-high temperature ceramic. Hence, I believe it should be published in Materials provided the authors, first, corrected the title. The present formulation looks too general. Secondly, a brief information on 3D printing (Section 5) is to be removed from the paper as it is not connected to the main part of the review.

Reviewer 2 Report

The present study highlights the review on the suspension dispersion and colloidal processing of ZrB2-based ultra high temperature ceramics. The manuscript is well written and covers the standard aspect of the application of the ceramic. Manuscript may be accepted. 

Reviewer 3 Report

This paper describes all aspects of powder processing routes for the manufacture of ZrB2 and ZrB2/SiC composites in theory as well as in practice. The comparison of the various methods, however, is only qualitative and not quantitative being supported by some numbers on porosity, density, mechanical properties, showing micro structures, etc. It is thus difficult to evaluate the various methods with each other.

In addition, there are some minor corrections to be made:

- please use subscripts: ZrB2, VVdW, Vsteric, Velectrostat, a, a2, etc.

  • p.3, line 100-102: sentence is incomplete, at least the verb is missing!
  • p.5, line 202 and 207: polyelectrolytes
  • p.6, line 229: ZrB2-SiC
  • p.8, line 294: AM-MBAM, please explain!
  • rename paragraph 4 in summary!

Reviewer 4 Report

The manuscript is a review of the colloidal processing of ZrB2 based ceramics.  The manuscript is well structured and well written and contains extensive references.  However, the manuscript suffers from several critical flaws:

  • Review has too much focus on general introductory material. The first four pages of the review cover material that is found in ceramics and colloidal processing textbooks.  While some introductory material should be included in a review article, this amount is excessive.  Further, the passage reads like a textbook source and does not well relate the material being discussed to the specific material system of the review.
  • Review does not present significant new findings or understanding from previous review. Unfortunately, the review does not offer significant new understanding over prior reviews.  For example:

Tallon, G.V. Franks, “Near-net-shaping of ultra-high temperature ceramics,” in Ultra-high temperature ceramics: materials for extreme environment applications, ed. W.G. Fahrenholz, E.J. Wuchina, W.E. Lee, Y. Zhou, John Wiley & Sons, Inc., Hoboken, NJ, 2014.

  • The review lacks depth and is not being selective about using high-quality references. For example, in Section 3.2. Tape Casting, the authors list over a dozen references, and do not expand the selected references beyond the reference authors having made tapes.
  • The authors have a tendency for definitive statements, several of which are easily falsifiable. One egregious example occurs at the beginning of the manuscript, where the authors state the “UHTCs are always hot-pressed at temperatures above 1800°C…”  This reviewer must assume this is an English language mistake, as otherwise it seemingly ignores over nearly 20 years of effort by the UHTC community and is at odds with the present review topic.

Reviewer 5 Report

The manuscript reviews the colloidal methods used for fabrication of ZrB2 ceramics and ZrB2 matrix composites. The text is written clearly. The manuscript starts with the description of particle-particle interactions principles. In the further part of the manuscript the brief description of the implementation of colloidal processing found in literature is given.

In my opinion, the subject is described in a very general way - starting with the fundamental principles of ceramic powder suspension description. For example, in chapter 2.1 the reactions occurring at the particle surface in water are given, but it is not said how it influences the ability to create a stable/not stable dispersion or how it influences the zeta potential.

The Hamaker model is given (which should simplified as a1=a2, as the Authors mentioned), but the Authors do not use it for calculations of particle-particle forces. The nature of Hamaker model and Hamaker constant is also not given. This implies a question, why was the model presented in the text.

Ionic dispersants mentioned in lines 162-173 are in the first meaning salts (not necessarily with polymeric chain). The ions adsorbed at the surface of the particles provide charge causing an increase of the potential in comparison with the solution. When the dispersant modifies the charge at the particle surface and contains polymeric chain that “physically” repulse other particles the “electrosteric” stabilization is than a more appropriate name.

Additionally, the text lacks literature references (generally throughout the whole text).

The text also should have been better edited (subscripts in chemical equations and symbols are missing).

The text is also not split in chapters in logical way (chapter 2.2 ends with no conclusion to be continued in 2.3).

The description of colloidal processing examples takes about 3 pages and the description of colloid chemistry – 4 pages, so the manuscript misses the subject of its title.

To sum up, the text is well organized and proves that there is scarce information on the colloidal processing of ZrB2, however, the description should be more insightful.

Round 2

Reviewer 5 Report

The revised manuscript is definitely better edited and the chapters are “closed” with some concluding remarks. However, the text of the manuscript did not undergo much modification.

For example, the 3D printing description is still very roughly presented. I understand, that the reader can refer for further information in the given reference, but a review paper should gather for the reader the knowledge.

Secondly, the theoretical background does not give any information on the ZrB2 material. The manuscript does not give information for example: why usually the aqueous slurries are used (which slurries are described in further part of the manuscript), how the corrosion influences the properties of the sintered material, what are the attraction forces/Hamaker constant in various liquids (“Hamaker 2” program could be of help in this case) most commonly used for preparation of ceramic slurries.

The manuscript has a potential to be a very helpful source of knowledge for scientists dealing with ZrB2 ceramics.  A review should present significant new findings or understanding from previous works and, ideally, lead author should have published extensively on the subject. Thus I strongly recommend to describe at least one aspect of colloidal processing of the Authors’ choice of ZrB2 more insightfully.
